# Protocol for a scoping review on technology use and sexual and gender minority youth and mental health

Kaitrin Doll[1]*, Shelley Craig[1], Yoonhee Lee[2], Toula Kourgiantakis[1], Eunjung Lee[1], Dane Dicesare[3], Ali Pearson[1], Tin Vo[4]

1 Factor-Inwentash Faculty of Social Work, University of Toronto, Toronto, ON, Canada, 2 User Services, Robarts Library, University of Toronto, Toronto, ON, Canada, 3 Department of Educational Studies Brock University, St. Catharines, ON, Canada, 4 Lyle S. Hallman Faculty of Social Work, Wilfrid Laurier University, Toronto, ON, Canada

* kaitrin.doll@mail.utoronto.ca

**Data Availability Statement:** All relevant data are within the paper and its Supporting Information files.

## Abstract

### Introduction

Research indicates that sexual and gender minority youth [SGMY] may engage more with information communication technologies [ICTs] more than their non-SGMY counterparts Craig SL et al. 2020. While scholarship generally explores youth's use of ICTs, there are gaps in scholarship that connect SGMY, their ICT engagement and influences to mental health. This scoping review will synthesize the literature that connects these core concepts in order to better understand the influence ITCs have on the mental health of SGMY and to develop a more fulsome understanding of this emerging area of literature.

### Methods and analysis

Following the scoping review framework of Arksey and O'Malley, the search will be conducted in the PsycINFO [Ovid interface, 1980-], MEDLINE [Ovid interface, 1948-], CINAHL [EBSCO interface, 1937-], Sociological Abstracts [ProQuest interface, 1952-], Social Services Abstracts [ProQuest interface, 1979-], and Scopus. Descriptive summaries and thematic analysis will summarize the articles that meet the inclusion criteria using an extraction table.

### Ethics and dissemination

The review outlined in this paper provides an overview of information that exists on the technology use of SGMY, ICTs and the interconnection with mental health. Results will be disseminated through peer reviewed journals and national and international conferences. As information collected for this paper as is retrieved from publicly available sources, ethics approval is not required.

**Funding:** The authors received no specific funding for this work.

**Competing interests:** The authors have declared that no competing interests exist.

## Introduction

There is limited scholarship specific to information and communication technologies (CTs) use and the mental health of sexual and gender minority youth (SGMY). For this review, sexual and gender minorities (SGM) include, but are not limited to, individuals who identify as lesbian, gay, bisexual, asexual, transgender, trans, Two-Spirit, queer, and/or intersex. Individuals with same-sex or-gender attractions or behaviors and those with a difference in sex development are also included in this definition. These groups also include individuals who may not use these terms to describe themselves, but their sexual orientation, gender identity or expression, or reproductive development doesn't fit into traditional or normative categories of sexual orientation, gender, or sex. SGMY experience mental health disparities because of minority stress connected to their SGM statuses [1, 2]. Specifically, minority stress theory demonstrates that SGM populations experience disproportionate mental health disparities because of the stigma, discrimination, and prejudice they experience that creates stressful environments that contribute to adverse mental and physical health outcomes [1, 3, 4]. Moreover, SGM populations are more likely to experience violence, victimization and a lack of social support that accentuates mental health disparities [5]. When compared to non SGM populations, SGM people experience higher rates of depression, anxiety, suicidal thoughts, self-harm, and substance use [4, 6]. Sexual minorities are documented to have suicide attempt rates between 20% and 42% [6]. Existing literature specific to SGMY demonstrates the population to be at disproportionate risk of negative experiences and outcomes, such as familial rejection [7], social exclusion [8], depression [9, 10] and low academic achievement [11]. Minority stress theories emphasise how experiences of marginalization, discrimination related to minority status can result in circumstances of isolation and victimization for SGMY at higher rates than their heterosexual peers [1, 12, 13]. Discrimination related to sexual orientation and gender identity can led to an accumulation of high overall levels of stress [14] and consequently SGMY are likely to endure emotional and behavioural consequences [15, 16]. Many SGMY internalize negative emotions and are at increased risk of developing depressive symptoms, anxiety, suicidal ideation and report lower levels of life satisfaction [12, 17–19]. Specifically, trans and gender diverse [TGD] people are seven times more likely to attempt suicide than the general population [20] and studies emphasize how TGD populations are exceptionally vulnerable for suicidality that results from internalized transphobia [21]. Wilson & Meyer [22] report that gender diverse people reported high rates of mental health concerns and 39% had attempted suicide, while James et al., [23] found that 82% of trans people report suicidal thoughts with 40% having attempted suicide. TGD youth are particularly vulnerable and experience stigma, violence and discrimination in their homes, schools and communities that exacerbate psychological distress [24]. Evidence demonstrates that suicidality is most prevalent among TGD youth with more than half the population noting experiences of suicidality [23–25].

The purpose of this review is to provide an overview of what information exists on the technology use of SGMY populations and any interconnection with mental health; specifically, this review will seek to find any peer reviewed literature that connects technology use and SGMY and mental health. For the review, information and communication technologies (ICTs) are offline (e.g., televisions, phone) and online (e.g., Internet, social media) technologies that facilitate communication and sharing of information, including mobile devices (e.g., mobile phones, tablets) [26, 27]. For this protocol, youth encompass people between the ages of (13–25), this includes terms such as young adults, adolescents, transition aged youth, some middle school students, high school students, college and university students and emerging adults. The Mental health Commission of Canada [28] distinguishes emerging adulthood (16–25) to describe a time of "significant growth and development" [p.5]. Others such as [29] differentiate

emerging adults from underage adolescents but note how this group is still in a period of "heightened instability" and transition, as such, emerging adults up to age 25 will be captured in this scoping review. Finally, mental illness/disorder refers to perceptions, feelings and disturbances in thoughts that cause distress and impair functioning [30]. This paper uses the term mental health broadly to capture mental disorders/illness but also emotional, psychological and social well-being to capture a range of mental health experiences.

"Zillenials" those born between 1995 and 2010 [31 p55] and are the first generation who have access to ICTs for the entirety of their lives [32, 33] and ICT use is intrinsic to youth lives and culture [34]. Scholarship on youth and ICTs tends to emphasise the negative impacts and risks of ICTs on youth focusing on topics such as smartphone addiction [35], online bullying, violence, and ease of access to problematic content and pornography [36]. This emphasis contributes to a negative perspective of youths' ICTs use a supports moral panic that problematizes youth's access and usage of ICTs [34]. While scholarship generally explores youth's use of ICTs, there is limited scholarship that speaks to the nuanced experiences of SGMY and their ICT engagement and how these experiences may differ from their non-SGMY peers. This lack of specificity is significant especially considering research indicates that SGMY may engage more with ICTs then their non-SGMY counterparts [37]. In a study conducted on the ICTs usage of SGMY in Canada and the US, results demonstrate that SGMY are heavy users of ITCs, with 47% typically spending more than five hours a day online using 2–3 devices [26]. Only recently has scholarship begun to emphasize the connection between SGMY identity development [24, 38] and their experiences with ICTs; increasingly research shows how ICTs can be affirming spaces [24, 26, 32, 38–40] that facilitate resilience [41, 42] and contribute to wellbeing for SGMY offline [32, 37].

Scholarship on SGMY and ICTs has revealed that SGMY are using ICTs as platforms to connect those who share similar identities [26, 32, 38, 41], to experiment with their SGMY identities [32], to come out [43], to receive support and provide support [24, 44, 45] and to participate in SGMY community where they feel validated in their identity [46, 47]. To date there has yet to be a scoping review that summarizes literature on SGMY, their ICT use and any connection to mental health or wellness. This review will bridge this gap and provide a comprehensive review of scholarship that exists around SGMY, their use of ICTs, and relevance to mental health to identify any gaps in scholarship and areas for future research.

## Study objectives

The overall study aim is to locate and summarize literature on sexual and gender minority youth and their use of information communication technologies and how this influences mental health. The objectives of this protocol are to (1) document, describe and summarize any literature that exists on sexual and gender minority youth populations and their use of ICTs; (2) demonstrate gaps in literature and areas for further exploration; (3) to provide recommendations for new research, practices and policies and areas for broader implications of any findings; and (4) to better understand the influence ICTs have on the mental health of SGMY.

## Methods and analysis

To map and explore the existing literature on the role of information communication technologies on the mental health of sexual and gender minority youth, we will conduct a scoping review. This scoping review will follow the methods and framework outlined by Arksey and O'Malley [48], which consists of five phases: (1) identifying the research question (2) identifying relevant studies, (3) study selection, (4) charting the data and (5) collating, summarizing, and reporting the results. Arksey and O'Malley [2005] point to the strength of scoping review

methodology, in allowing researchers to systematically explore existing relevant literature, identify key concepts and themes, and uncover gaps in the literature. This protocol was developed using the Preferred Reporting Items for Systematic Review and Meta-Analysis Protocols [PRISMA-P] checklist [49]. See S2 Appendix. The scoping review will adhere to the Preferred Reporting Items for Systematic review and Meta-Analysis protocols [PRISMA-P] guidelines [49]. The scoping review will be conducted by seven researchers, three Factor Inwentash Faculty of Social Work faculty members at University of Toronto [SC, EL, TK], a Lyle S. Hallman Faculty of Social Work researcher at Wilfrid Laurier University (TV), a Department of Educational Studies faculty member at Brock University (DD), and two doctoral students from Factor Inwentash Faculty of Social Work (KD, AP). The research protocol was developed by a doctoral candidate (KD) and full professor (SC) and the librarian for social work (YL) at University of Toronto developed the search strategy methods. The research team approved the research protocol and confirmed it aligned with the objectives of the scoping review.

## Step 1: Identifying the research question

As noted in the preliminary review of the literature, there is emerging literature that connects SGMY, ICTs and mental health. Based on the objectives, the research questions were developed by (KD, SC and YL) and discussed with the full research team for approval. As this scoping review aims to discover more about the relationship between the three core concepts (SGMY, ICTs and mental health) the following research questions were developed: (1) how does literature describe the relationship between SGMY, ICTs and mental health (2) what are the gaps and areas for future research identified in the literature on SGMY, ICTs and mental health? (3) what are the recommendations for new research, practices and policies, and areas for broader implications of any findings? (4) how does the literature describe ICT use and how it's associated with mental health of SGMY?

## Step 2: Identifying relevant studies

In collaboration with the research team, the social work librarian (YL) will develop a comprehensive search of published literature. The search strategies will be developed using both keywords and controlled vocabulary, related to the concepts of SGMY, ICTs, and mental health. We will search the following electronic databases: PsycINFO (Ovid interface, 1980-), MEDLINE (Ovid interface, 1948-), CINAHL (EBSCO interface, 1937-), Sociological Abstracts (ProQuest interface, 1952-), Social Services Abstracts (ProQuest interface, 1979-), and Scopus. These databases were chosen to ensure a comprehensive search of relevant literature through psychology, science, health and sociology databases. In addition to database searching, the reference lists of articles that meet inclusion criteria will be scanned for additional sources to ensure literature saturation.

A pilot search strategy was developed in PsycINFO, using text words and APA Thesaurus Terms, related to sexual and gender minority youth, information communication technologies, and mental health (see S1 Appendix). The search strategy was further refined by the social work librarian (YL) with input and consultation with the rest of the research team. The draft search strategy was peer-reviewed by a second librarian according to the Peer Review of Electronic Search Strategies (PRESS) guidelines, which led to further revisions and refinement [50]. After finalization, the PsycINFO search strategy will be translated to the remaining databases. Searches will be limited to English and French language publications published on or after 2010. We included studies published in 2010 as this period marks when social media became highly utilized by youth and studies were beginning to establish the link between technology use and mental health and SGMY identities. Through a preliminary review of articles,

we observed that there were limited instances of such connections being explored prior to 2010. Our focus on studies since 2010 allowed us to explore the most relevant to generate a comprehensive and contemporary overview of these relationships.

### Step 3: Study selection

Literature selected for the scoping review will be based on the following inclusion criteria: (1) written in English or French; (2) published on or after 2010; (3) focuses on participants between 13–25 (4) includes all three key terms: ICTs, sexual gender minorities youth and mental health and; (5) empirical studies, teaching papers, different types of reviews, conceptual and theoretical papers in peer reviewed journals are eligible for this study.

Articles deemed to meet the inclusion criteria will be retrieved from the databases and imported into reference management software called Zotero. The RIS file will also be uploaded into Covidence, a web-based software used for scoping reviews that supports screening and study selection, where all reviewers can access the results [49]. In advance of the article screening, the service user librarian (YL) will ensure there is no duplication of articles. A two-stage review strategy will be used to screen articles. In the first stage, reviewers (KD, DD, AP) will use Covidence to independently screen articles based-on titles, keywords and abstracts using the scoping review inclusion criteria [51]. In stage two, four team members (SC, EL, TG, KD) will independently complete a full text-review of articles chosen in stage one. If reviewers do not agree on the inclusion of an article all the reviewers will discuss to determine whether the article should be included, first author of the scoping review (SC) will resolve any discrepancies.

### Step 4: Charting the data

After completing the database searches, the articles that meet the inclusion criteria will be analyzed to extract key information that will be systematically categorized and organized. The scoping review objectives and research questions provide a framework for the charting categories to capture relevant information from literature included in the scoping review. Based on the articles that pass the inclusion criteria, variables from literature eligible for inclusion in the review will be collected. A tracking sheet will be developed to record important categories from selected papers. The tracking sheet will include: (1) date of search; (2) search string; (3) authors; (4) study location; (5) discipline; (6) method; (7) type of publication; (8) Specifics of SGMY groups identified in the study; (9) concepts discussed; (10) research questions/study aims/purpose; (11) key findings; (12) interventions used; (13) types of ICTs identified in the study; (14) how mental health is addressed in the article (15) how mental health and ICTs are linked (16) recommendations for future research; (16) broader application of research for policy and practice; and (17) limitations as identified by authors.

Two members of the team will pilot the data extraction template to ensure that the framework collects the appropriate information and can be consistently applied; categories will be modified if necessary [52]. Researchers will also use a table to track any changes to the keywords or search strings that may be modified in the search process using different databases. This table will include: (1) date of the search; (2) the search string; and (3) the number of articles generated.

### Step 5: Identification, synthesis and report of study findings

Descriptive summaries and thematic analysis will summarize the results from the extraction table to present an overview of the reviewed literature [53]. The analysis will highlight common characteristics and key themes from each article using a consistent approach, enabling

comparisons between articles and the identification of recommendations, gaps, limitations, and areas for further literature development. A report will be produced summarizing the review findings, common themes, identifying gaps and areas for further research and noting other findings as related to the scoping review objectives. The report will provide information about the current research on SGMY, ICTs and mental health; moreover, summarizing and disseminating research findings can contribute to more nuanced understandings of the connection between the three core concepts to provide practitioners, clinicians, researchers and consumers with the most recent academic literature on the topic.

## Discussion

Given the vulnerabilities of SGMY related to minority stress and the well-documented use of ICTs among youth, it is critical to explore any significance between ICT use and mental health. Our review aims to explore evidence of what information exists on the technology use among SGMY populations and any interconnection with mental health. This review will help to identify gaps in research and provide new directions for approaches to supporting SGMY mental health and engagement with ICTs.

The proposed scoping review will allow the research team to carry out a systematic inquiry in this important area and provide an overview of the evidence pertinent to SGMY, ICTs, and mental health. Information collected for this review has been retrieved from publicly available sources, therefore ethics approval is not required. The review excludes grey literature to ensure that the included studies are evidence based and published in peer reviewed journals. Results will be disseminated through peer reviewed journals and national and international conferences. Amendments to this protocol, if any, will be listed in the final review publication.

## Supporting information

**S1 Appendix. Search strategy for PsycINFO.**
(DOCX)

**S2 Appendix. PRISMA-P (Preferred Reporting Items for Systematic review and Meta-Analysis Protocols) 2015 checklist: Recommended items to address in a systematic review protocol*.**
(DOCX)

## Author Contributions

**Conceptualization:** Kaitrin Doll, Shelley Craig, Yoonhee Lee, Eunjung Lee.

**Data curation:** Ali Pearson, Tin Vo.

**Formal analysis:** Dane Dicesare.

**Methodology:** Yoonhee Lee, Toula Kourgiantakis.

**Resources:** Ali Pearson, Tin Vo.

**Writing – original draft:** Kaitrin Doll.

**Writing – review & editing:** Kaitrin Doll, Shelley Craig, Yoonhee Lee, Toula Kourgiantakis, Dane Dicesare.

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
