## [Decision Letter · Decision Letter 0]

23 Aug 2023

PONE-D-23-06193Protocol for a Scoping Review on Technology Use and Sexual and Gender Minority Youth and Mental HealthPLOS ONE

Dear Dr. Doll,

Thank you for submitting your manuscript to PLOS ONE. After careful consideration, we feel that it has merit but does not fully meet PLOS ONE’s publication criteria as it currently stands. Therefore, we invite you to submit a revised version of the manuscript that addresses the points raised during the review process.

We look forward to receiving your revised manuscript.

Kind regards,

Vincenzo De Luca

Academic Editor

PLOS ONE

Journal Requirements:

Reviewers' comments:

Reviewer's Responses to Questions

**Comments to the Author**

1. Does the manuscript provide a valid rationale for the proposed study, with clearly identified and justified research questions?

Reviewer #1: Yes

2. Is the protocol technically sound and planned in a manner that will lead to a meaningful outcome and allow testing the stated hypotheses?

Reviewer #1: Partly

3. Is the methodology feasible and described in sufficient detail to allow the work to be replicable?

Reviewer #1: Yes

4. Have the authors described where all data underlying the findings will be made available when the study is complete?

Reviewer #1: No

5. Is the manuscript presented in an intelligible fashion and written in standard English?

Reviewer #1: Yes

6. Review Comments to the Author

You may also provide optional suggestions and comments to authors that they might find helpful in planning their study.

Reviewer #1: This protocol proposes a scoping review to understand the current state of literature exploring the relationship between sexual and gender minority youth, technology and mental health. This is a highly interesting and important topic and I would be very interested to see the results of such a review. I had a few thoughts/comments which I hope are useful.

Does feel a shame not to assess the quality to some degree. Part of a scoping review is presumably assessing what still needs to be conducted. It is hard to deem where the gaps are without knowing the quality of the evidence? For example, there might be an association which has already been explored by a large number of studies, but all of them are cross-sectional and don’t control for confounding – this would therefore be a big gap in the literature in terms of study quality that would be important to know about. Even if it isn’t judged in a formal way, I think it would be very useful to comment on the quality of the studies.

I think it would be good to move the definition of ‘sexual and gender minority youth’ further up in the introduction (ideally when the term is first introduced). Until I reached the definition, I was not sure whether ‘sexual’ included people with intersex conditions.

SGMY is an incredibly broad umbrella. I would hypothesise that the relationship between technology use and mental health might be different for someone who is gay v. someone who is transgender v. someone with Turner syndrome, for example. Perhaps some justification is needed for why looking at all of these groups at the same time? Do you plan to synthesise/summarise the results for the different groups separately?

“These populations also encompass those who do not self-identify with one of these terms but whose sexual orientation, gender identity or expression, or reproductive development is characterized by non-binary constructs of sexual orientation, gender, and/or sex.” Not sure that I follow who this is supposed to capture. People who accept that sexual orientation is not a binary, but they themselves are heterosexual? I am sure this isn’t what you mean, but this is how I would interpret that sentence so maybe you could rephrase/clarify?

Why only studies conducted after 2010?

7. PLOS authors have the option to publish the peer review history of their article (what does this mean?). If published, this will include your full peer review and any attached files.

Reviewer #1: No

---

## [Author Response · Author response to Decision Letter 0]

31 Aug 2023

Letter to Reviewers

Regarding: Protocol for a Scoping Review on Technology Use and Sexual and Gender Minority Youth and Mental Health 

We are grateful for your thoughtful feedback on the protocol. We have worked to integrate your feedback and believe it has strengthened the manuscript. Below, we address your comments and provide further clarification.

Comments to the Author

1. Does the manuscript provide a valid rationale for the proposed study, with clearly identified and justified research questions?

Reviewer #1: Yes

Response to reviewer: Thank you.

2. Is the protocol technically sound and planned in a manner that will lead to a meaningful outcome and allow testing the stated hypotheses?

Reviewer #1: Partly

Response to reviewer: We appreciate this feedback. We do not explicitly plan to undertake any statistical analysis in this scoping review; however, we have made some adjustments to the methods to better respond to this feedback. The analysis, and described in paper/protocol is the gold standard for scoping reviews as outline by Arksey and O'Malley (2005). Specially the analysis will highlight common characteristics and key themes from each article using a consistent approach, enabling comparisons between articles and the identification of recommendations, gaps, limitations, and areas for further literature development. Please see page 14

3. Is the methodology feasible and described in sufficient detail to allow the work to be replicable?

Reviewer #1: Yes

4. Have the authors described where all data underlying the findings will be made available when the study is complete?

Reviewer #1: No

Response to reviewer: Once the study is completed we will post the relevant data on the open access repository affiliated with our academic institution and plan to include as much as possible in the initial manuscript.

5. Is the manuscript presented in an intelligible fashion and written in standard English?

Reviewer #1: Yes

6. Review Comments to the Author

Reviewer #1: This protocol proposes a scoping review to understand the current state of literature exploring the relationship between sexual and gender minority youth, technology and mental health. This is a highly interesting and important topic and I would be very interested to see the results of such a review. I had a few thoughts/comments which I hope are useful.

Does feel a shame not to assess the quality to some degree. Part of a scoping review is presumably assessing what still needs to be conducted. It is hard to deem where the gaps are without knowing the quality of the evidence? For example, there might be an association which has already been explored by a large number of studies, but all of them are cross-sectional and don’t control for confounding – this would therefore be a big gap in the literature in terms of study quality that would be important to know about. Even if it isn’t judged in a formal way, I think it would be very useful to comment on the quality of the studies.

Response to reviewer: We appreciate this feedback. Although the scoping review protocol that we plan to utilize (Arksey, O'Malley, 2005) does not identify strategies to evaluate quality, we will note informally and ultimately include any information we deem helpful to highlight quality issues. Generally, scoping reviews do not involve the evaluation of study quality as a defining characteristic (Rumrill, Fitzgerald & Merchant, 2010; Brien et al., 2010) Rather, scoping studies primarily focus on mapping the breadth and depth of evidence within a field, presenting an overview of existing research (Davis, Drey & Gould, 2009). This approach differentiates scoping studies from both systematic reviews and narrative/literature reviews, as the scoping process emphasizes analytical reinterpretation of the literature rather than quality assessment (Davis, Drey & Gould, 2009). The strength of a scoping review is its breadth which permits us to map the literature available without limitations and provide a broad portrait. That said, we agree that while we will not appraise the quality of the articles, we will discuss their characteristics, limitations, and gaps. We have made the following change in our manuscript:

Strengths and limitations of this study

• Studies included in the review will not be appraised for quality, however gaps and limitations of selected studies will be discussed. P.3

I think it would be good to move the definition of ‘sexual and gender minority youth’ further up in the introduction (ideally when the term is first introduced). Until I reached the definition, I was not sure whether ‘sexual’ included people with intersex conditions.

Response to reviewer: The definition has been moved to the introduction section see page 4. 

SGMY is an incredibly broad umbrella. I would hypothesise that the relationship between technology use and mental health might be different for someone who is gay v. someone who is transgender v. someone with Turner syndrome, for example. Perhaps some justification is needed for why looking at all of these groups at the same time? Do you plan to synthesise/summarise the results for the different groups separately?

Response to reviewer: We appreciate your insight into the diversity encompassed within the SGMY umbrella. While we acknowledge the broad nature of this category, we aimed to conduct a comprehensive search considering the niche nature of this area. It's worth noting that numerous studies concentrate on SGMY as a collective population, often presenting specific identities within their demographic data. In our analysis, step 4 charting the data, number (8) states “(8) Specifics of SGMY groups identified in the study” (see page 13) we plan to delineate studies that particularly emphasize distinct SGMY subgroups. This approach will allow us to explore potential variations in the relationship between technology use and mental health across different groups within the SGMY spectrum. 

“These populations also encompass those who do not self-identify with one of these terms but whose sexual orientation, gender identity or expression, or reproductive development is characterized by non-binary constructs of sexual orientation, gender, and/or sex.” Not sure that I follow who this is supposed to capture. People who accept that sexual orientation is not a binary, but they themselves are heterosexual? I am sure this isn’t what you mean, but this is how I would interpret that sentence so maybe you could rephrase/clarify?

 Response to reviewer: The sentence was rephrased to “These groups also include individuals who may not use these terms to describe themselves, but their sexual orientation, gender identity or expression, or reproductive development doesn't fit into traditional or normative categories of sexual orientation, gender, or sex. See page 4. 

Why only studies conducted after 2010?

Response to reviewer: This section was added to Step 2: Identifying Relevant Studies: 

We included studies published in 2010 as this period marks when social media became highly utilized by youth and studies were beginning to establish the link between technology use and mental health and SGMY identities. Through a preliminary review of articles, we observed that there were limited instances of such connections being explored prior to 2010. Our focus on studies since 2010 allowed us to explore the most relevant to generate a comprehensive and contemporary overview of these relationships. See page 10.

---

## [Editor Report · Decision Letter 1]

1 Sep 2023

Protocol for a Scoping Review on Technology Use and Sexual and Gender Minority Youth and Mental Health

PONE-D-23-06193R1

Dear Dr. Doll,

We’re pleased to inform you that your manuscript has been judged scientifically suitable for publication and will be formally accepted for publication once it meets all outstanding technical requirements.

Kind regards,

Apurva kumar Pandya, PhD

Academic Editor

PLOS ONE

Additional Editor Comments (optional):

As comments are sufficiently addressed, this can be accepted.
---

## [Editor Report · Acceptance letter]

14 Sep 2023

PONE-D-23-06193R1 

Protocol for a Scoping Review on Technology Use and Sexual and Gender Minority Youth and Mental Health 

Dear Dr. Doll:

I'm pleased to inform you that your manuscript has been deemed suitable for publication in PLOS ONE. Congratulations! Your manuscript is now with our production department. 

Kind regards, 

on behalf of

Dr. Apurva kumar Pandya 

Academic Editor

PLOS ONE